# Review of Bubble Applications in Microrobotics: Propulsion, Manipulation, and Assembly

**DOI:** 10.3390/mi13071068

**Published:** 2022-07-04

**Authors:** Yuting Zhou, Liguo Dai, Niandong Jiao

**Affiliations:** 1State Key Laboratory of Robotics, Shenyang Institute of Automation, Chinese Academy of Sciences, Shenyang 110016, China; zhouyuting@sia.cn; 2Institutes for Robotics and Intelligent Manufacturing, Chinese Academy of Sciences, Shenyang 110016, China; 3University of Chinese Academy of Sciences, Beijing 100049, China; 4College of Mechanical and Electrical Engineering, Zhengzhou University of Light Industry, Zhengzhou 450002, China; dailiguo1989@163.com

**Keywords:** microbubbles, microrobots, micromanipulation, microassembly

## Abstract

In recent years, microbubbles have been widely used in the field of microrobots due to their unique properties. Microbubbles can be easily produced and used as power sources or tools of microrobots, and the bubbles can even serve as microrobots themselves. As a power source, bubbles can propel microrobots to swim in liquid under low-Reynolds-number conditions. As a manipulation tool, microbubbles can act as the micromanipulators of microrobots, allowing them to operate upon particles, cells, and organisms. As a microrobot, microbubbles can operate and assemble complex microparts in two- or three-dimensional spaces. This review provides a comprehensive overview of bubble applications in microrobotics including propulsion, micromanipulation, and microassembly. First, we introduce the diverse bubble generation and control methods. Then, we review and discuss how bubbles can play a role in microrobotics via three functions: propulsion, manipulation, and assembly. Finally, by highlighting the advantages and current challenges of this progress, we discuss the prospects of microbubbles in microrobotics.

## 1. Introduction

Bubbles are attractive, magical, multifunctional, and ubiquitous in industrial production and our daily life. They have a variety of physical properties including large specific surface area, low density, and surface hydrophobicity. Applications based on bubbles have received extensive attention and research in recent decades [1,2]. Depending on their sizes, bubbles can be categorized into macrobubbles, microbubbles, and nanobubbles. Macrobubbles are 2–5 mm in diameter. At the microscopic scale, bubbles can be classified into microbubbles (diameters of 1–100 μm) and nanobubbles (diameters below 1 μm) [3]. Following the development of microfluidics and microrobotics in recent years, micro/nanoscale bubbles have been seen as an emerging tool for solving numerous challenges in various lab-on-a-chip (LOC) applications, and they are gaining increasing attention from researchers.

In the micro/nano research field, bubble-based applications have attracted increasing attention because of their simplicity, controllability, and biocompatibility. They can be flexibly integrated with different microfluidic devices or microrobots. For example, in microfluidics, bubbles can be remotely excited by an acoustic field to act as micromixers [4,5], micropumps [6,7], or microvalves [8,9], and they can operate upon particles and cells [10,11]. In addition, in microrobotics, bubbles can act as the manipulating or transmission components of microrobots [12,13], and the bubbles themselves can act as microrobots [14]. However, the bubble generation methods and their applications in the microrobotics field (e.g., propulsion, micromanipulation, and microassembly) have not been classified and summarized in detail.

In this review, we introduce and discuss the propulsion, manipulation, and assembly capabilities of bubbles in microrobotics. We demonstrate the importance, flexibility, and versatility of bubbles, as described in typical research papers. A schematic diagram of the generation and control methods of bubbles and their roles in microrobotics is shown in Figure 1. In Section 2, we introduce the methods of bubble generation (chemical reaction, direct acquisition, and optothermal effect) and control (acoustic oscillation, optothermal effect, and electrowetting-on-dielectric (EWOD) technology). In Section 3, we discuss how bubbles can be used as propulsion mechanisms for microrobots (e.g., tubular micromotors, Janus particles, and self-propelled micromachines). In Section 4, we demonstrate how bubbles act as the tools of microrobots and can be used for micromanipulation or transmission. In Section 5, we introduce bubbles as microrobots to achieve micromanipulation and microassembly in two-dimensional (2D) and three-dimensional (3D) spaces. Finally, we summarize the current limitations of bubbles in microrobotics and discuss future developments.

## 2. Generation and Control of Bubbles

There are many methods to produce and control bubbles. In this section, we review several of them. The bubble generation methods include chemical reactions, direct acquisition, and the optothermal effect. Bubble control methods include acoustic oscillation, optothermal effect, and EWOD technology. These methods can be effectively combined to develop diverse applications based on bubbles. For example, the direct acquisition of air to generate bubbles can be combined with acoustic oscillation to fabricate microswimmers via facile production processes; these offer good propulsion performances. The combination of EWOD technology and acoustic excitation can realize the movement and operational abilities of bubble microrobots.

### 2.1. Chemical Reaction

Here, the principle of bubble formation via chemical reactions and its application in self-propelled microrobots are introduced. Many chemical reactions produce gas; furthermore, when the chemical reaction occurs in a liquid environment, gas agglomerates can form gas bubbles. Among the bubble-generating chemical reactions, three types are effective and widely applied: the decomposition of hydrogen peroxide (H_2_O_2_), water (H_2_O) electrolysis, and metal oxidation. For example, H_2_O_2_ decomposition generates oxygen (O_2_) to form bubbles via
(1)$$H_{2}O_{2}\overset{\mathrm{catalyst}}{\to }H_{2}O+O_{2}↑.$$

H_2_O_2_ decomposes slowly; hence, gold (Au), platinum (Pt), and titanium dioxide (TiO_2_) are usually used as catalysts [15,16]. This reaction is often used to drive the movement of tubular micromotors and bimetal nanorods [17]. For example, in one study, the H_2_O_2_ solution was pumped into the front of a catalytic microtubular jet engine, and bubbles were generated at the other end of the larger opening, pushing the tube unidirectionally upon leaving (Figure 2a) [18].

Electrolytic water produces microbubbles of hydrogen (H_2_) and O_2_:(2)$$2H_{2}O\overset{\mathrm{electricity}}{\to }H_{2}↑+O_{2}↑.$$

The location, shrinkage, and parameters of the bubbles generated via water electrolysis can be easily controlled, and the reaction environment is non-toxic; hence, bubbles can be generated easily with high efficiency [19]. For example, programmable patterns of H_2_ and O_2_ bubbles can be electrochemically generated as required on the gold and copper electrodes of a complementary metal-oxide-semiconductor (CMOS) chip, which consists of 100 × 100 electronically addressable pixels. The size of the bubble can be determined via the electrolysis produced by the controlled current at each pixel (Figure 2b) [20].

Moreover, metal oxidation produces bubbles. Ideal Janus particles can be prepared by splitting water into hydrogen bubbles using biocompatible and environmentally friendly active metals such as magnesium (Mg), aluminum (Al), iron (Fe), zinc (Zn), and so on [21,22]. A transient self-destroyed micromotor fabricated from Mg/zinc oxide (ZnO) can achieve rapid propulsion via the reaction between Mg and water (Figure 2c) [23]. The three types of chemical reaction methods exhibit unique characteristics: electrolytic water methods can produce relatively long-life bubbles with low energy consumption, which makes them suitable in microfluidics roles; however, the equipment required is complex. In applications based on the other two chemical reactions, the service life of the bubbles is not a problem. Instead, these continuous bubbles realize the promotion of tubular micromotors and Janus particles. Bubble propulsion micro/nanomotors based on these two reactions have become the most popular because of their fast and highly dynamic movement, as a result, they have been the subject of extensive research.

**Figure 2 micromachines-13-01068-f002:**
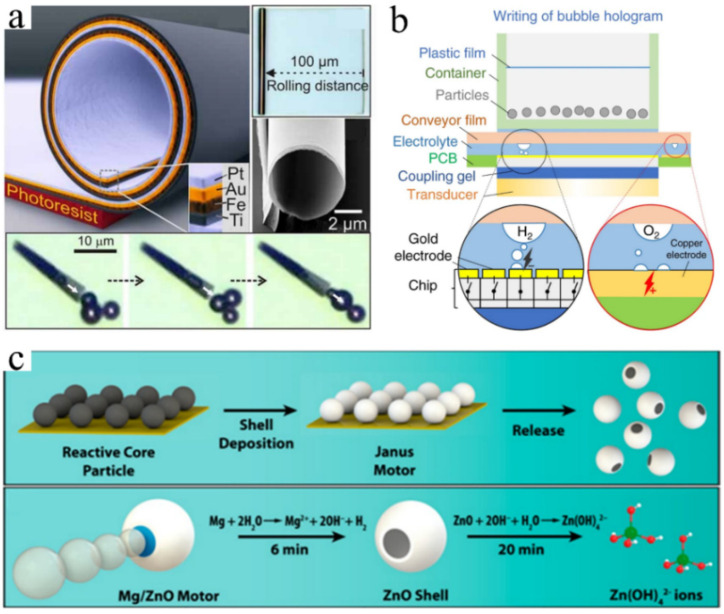
(**a**) Actuation of the tubular micromotor via bubbles generated from the H_2_O_2_ reaction. Adapted from Solovev et al. [18] with permission from John Wiley and Sons, Copyright 2009. (**b**) The electrochemical generation of programmable on-demand H_2_ and O_2_ bubbles at the gold and copper electrodes. Adapted from Ma et al. [20] with permission under the terms of the CC BY 4.0 License, Copyright 2020. (**c**) Janus micromotor propelled by bubbles generated via the chemical reaction of Mg with water. Adapted from Chen et al. [23] with permission from the American Chemical Society, Copyright 2016.

### 2.2. Direct Acquisition

The direct acquisition method uses the incompatibility of air (or other gases) with liquids to form bubbles directly [24]. These methods can be divided into active and passive methods. The active method, also known as the injection method, involves injecting an appropriate amount of nitrogen into the liquid of a microfluidic device to generate bubbles directly therein (Figure 3a) [25]. Passive trapping involves the use of hydrophobic materials to trap bubbles. Multiphase fluid systems tend to minimize their total surface energy, and higher contact angles and lower wettability are conducive to air trapping [26]. Therefore, hydrophobic surfaces can be used to capture bubbles floating nearby with relatively low surface tension [27], and realize controlled and directional bubble transport [28,29]. When grooves or other cavity structures are present inside the micropipe, air will be trapped because of the surface tension after water or other liquids are poured into the pipe and forced to become bubbles [30,31,32]. In addition, the hydrophobicity of the material facilitates the formation of bubbles in the cavity after a semi-enclosed cavity structure is placed in water (Figure 3b) [33,34]. Active and passive methods have their own advantages and disadvantages, respectively. The injection method can flexibly control the generation position and sizes of the bubbles as needed. However, due to the influence of manual control and environmental variables, it is difficult to obtain batch bubbles of the same size. The passive method can quickly and conveniently produce a large number of bubbles with specifications corresponding to the shape and size of the cavity. In general, direct acquisition represents a simple and low-cost bubble generation method.

### 2.3. Optothermal Effect

In daily life, air bubbles occur in cold water during heating because the solubility of air decreases when the temperature increases, and gases originally dissolved in the water are released. This principle can also be used to generate bubbles at the micro/nanoscale levels. Bubbles can be generated and controlled optically via the optothermal effect, which converts light energy into heat energy. Due to the thermal flow, optothermal bubbles have an adsorptive capacity and can be directly used to manipulate micro-objects. The generation and application of optothermal bubbles can be classified into two types: (1) Microbubbles are generated on the solid surface of heat-absorbing materials, which are often described as micromanipulation microrobots and are introduced in this subsection. (2) Micro or nanobubbles generated at the interface of the colloidal suspension and at a plasmonic substrate via plasmon-enhanced photothermal effects; this can pattern colloidal particles on substrates, referred to as bubble-pen lithography (BPL). Zheng et al. proved that nanoparticles could be written directly onto Ag films using optothermally generated surface bubbles (OGSB), and they realized single particle modes and particle combinations with different resolutions and structures [35]. This technology has attracted extensive attention from scholars in recent years, and the readers are invited to refer to the following references for more detailed discussions on this topic [36,37,38,39,40,41,42].

The optothermal effect is usually used to convert light energy into heat energy, and microbubbles are generated at the interface between heat-absorbing materials (e.g., metal, amorphous silicon, indium tin oxide, or their combination) and liquids [14]. The experimental device and movement of the optothermal bubble are shown in Figure 4a. Due to the different types of heat-absorbing materials, the applicable laser wavelength types also differ, primarily including near-infrared and ultraviolet light. The generation and size control of optothermal bubbles are related to the light absorptivity and laser spot density, and the light absorptivity is closely related to the material and thickness of the absorption layer as well as the laser wavelength. Because the irradiation range of the laser after focusing is very small, bubbles can be generated and controlled using the thermal gradient field generated by the heat-absorbing material after light energy absorption. Therefore, the optothermal method can accurately control the generation positions of bubbles; furthermore, when the position of the spot changes, the bubbles follow the spot [43].

The laser light exposed onto the endothermic layer (after passing through the optical appendix and focusing objective) is circular with a Gaussian power density distribution, and the input thermal energy Q across the endothermic layer in unit time is [44,45]
(3)$$Q=\frac{P\eta_{A}\eta_{T}}{2\pi R^{2}}exp\left(-\frac{r^{2}}{2R^{2}}\right),$$
where P is the input power of the laser; R is the radius of the spot; r is the distance from the center of the spot; $\eta_{A}$ is the laser’s photothermal efficiency; and $\eta_{T}$ is its optothermal conversion efficiency. When a laser beam irradiates the chip, the heat-absorbing layer converts the light energy into heat energy; this is transferred through the chip and into the water solution. When the laser is temporarily turned off, the temperature decreases accordingly. In the fluid and solid regions, the temperature distribution can be determined using Fourier’s law [46,47], expressed as:(4)$$\rho C_{p}\frac{\partial T}{\partial t}+\nabla \cdot \left(-k_{T}\nabla T\right)+\rho C_{p}u\cdot \nabla T=Q,$$
where ρ is the density; $C_{p}$ is the heat capacity; T is the temperature; t is the time; $k_{T}$ is the coefficient of heat conduction; u is the liquid velocity; and Q is the input power density. Temperature is the main factor affecting the generation of microbubbles. In the fluid and bubble areas, heat transfer causes flow, and the flow velocity can be described by the Navier–Stokes equations [48,49], as:(5)$$\rho \frac{\partial u}{\partial t}+\rho \left(u\cdot \nabla \right)=\nabla \cdot \left[-pI+\mu \left(\nabla u+{\left(\nabla u\right)}^{T}\right)\right]+\rho g,$$
(6)ρ∇⋅u=0,
where p is the pressure; I is the identity matrix; u is the viscosity; and g is the gravitational acceleration. A temperature gradient and convection flow pattern can be formed around the optothermal bubbles (Figure 4b) [50]. The temperature decreases along the radial direction via convective cooling along the top and bottom surfaces. This temperature gradient causes a corresponding convective flow that forms a clockwise flow pattern near the bubble–liquid interface. This microscale circulation is caused by the Marangoni effect [51,52]. The velocity field between the liquid layers (caused by the thermal Marangoni effect) can be described as [53]
(7)$$\eta \left(\frac{\partial \mu }{\partial n}\right)=\gamma_{T}\left(\frac{\partial T}{\partial t}\right),$$
where η is the dynamic viscosity; μ is the tangential component of the fluid velocity vector at the liquid–air interface; and n and t are the unit vectors of the normal and tangential directions of the interface, respectively. $\gamma_{T}=\partial \gamma /\partial T$ is the derivative of the surface tension γ with respect to temperature. The Marangoni convection and surface tension of the optothermal bubble exert forces on the particles suspended in solution to achieve particle manipulation [54], cell deformations [55], cell perforation, and lysis [56,57]. Optothermal bubbles can be generated and moved flexibly to any position on the surface of a 2D chip. Their size can be easily controlled, and they can attract and operate micro-objects without the assistance of other physical fields. However, the operation and applications of optothermal bubbles are limited by the 2D plane. Realizing multiple optothermal bubble clusters requires complex and expensive equipment; however, the application value must be further developed.

**Figure 4 micromachines-13-01068-f004:**
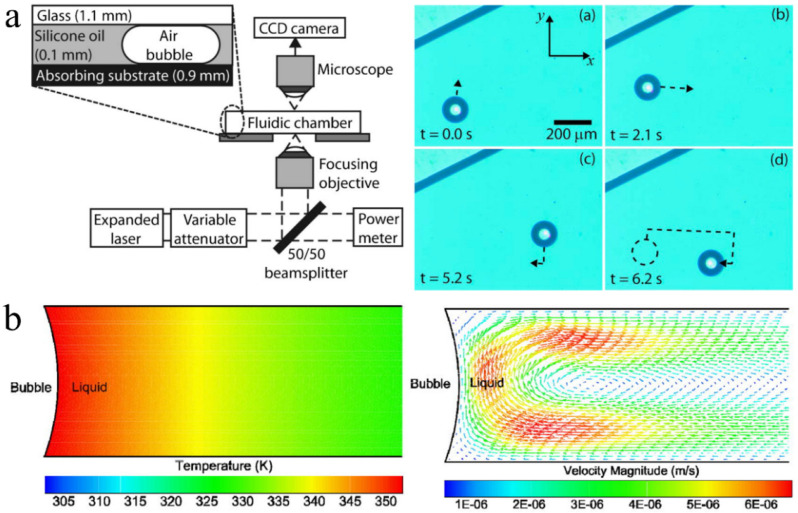
The optothermal effect used to generate and control bubbles. (**a**) Experimental device and the movement of optothermal bubbles. Adapted from Ohta et al. [14] with permission from AIP Publishing, Copyright 2007. (**b**) The temperature distribution and convective flow pattern around an optothermally generated bubble. Adapted from Zhao et al. [50] with permission from The Royal Society of Chemistry, Copyright 2014.

### 2.4. Acoustic Oscillation

The bubbles excited by acoustic waves and the bubble-oscillating mechanism used in operation and propulsion are described in this section. Microbubbles can be used in large-scale propulsion and operation because they resonate under the action of acoustic waves and oscillate the gas–liquid interface, which changes the flow direction and speed of the microfluids. Oscillating bubbles include inertial and non-inertial cavitation [58]. Inertial cavitation occurs when the oscillation amplitude of a bubble is sufficiently high and exceeds a certain threshold; then, the bubble expands and shrinks sharply or even collapses. This reaction can be used in biological applications such as cell sorting and cell lysis [59,60]. Most oscillating bubbles in microfluids are stable, non-inertial cavitated bubbles. Oscillating bubbles have natural frequencies, and when they have the same frequency as the excitation acoustic field, they reach a maximum vibration amplitude [1], expressed as
(8)$$f_{0}=\frac{1}{2\pi }\sqrt{\frac{1}{\rho }\left[3\kappa \left(P_{0}+\frac{2\sigma }{R}-P_{V}\right)-\left(\frac{2\sigma }{R}-P_{V}+\frac{4\mu^{2}}{\rho R_{0}}\right)\right]}\frac{1}{R_{0}},$$
where μ is the dynamic viscosity of the liquid; $R_{0}$ and R are the initial and instantaneous radii of the bubbles, respectively; $P_{0}$ is the constant far-field pressure; σ is the surface tension at the gas–liquid interface of the bubble; $P_{V}$ is the vapor pressure; and κ is the polytropic index. In addition, the $f_{0}$ of the bubbles in the tube or microchannel is
(9)$$f_{0}=\frac{1}{2\pi }\sqrt{\frac{\kappa P_{0}}{\rho L_{0}L_{B}},}$$
where $L_{0}$ and $L_{B}$ are the lengths of the liquid in the tube and the bubble column, respectively. Acoustically vibrating bubbles usually exhibit two typical oscillation modes: (1) volumetric or (2) reciprocal along the axis [61,62]. Under the excitation of acoustic waves, the particles close to the bubbles are primarily affected by the acoustic streaming force (estimated by Stokes drag force), that is, the viscous resistance caused by the streamline direction of the micro flow field, and the secondary acoustic radiation force (also known as the Bjerknes force) caused by the scattering effect of the bubble on the incident acoustic wave. The dominant force can be calculated by the ratio of magnitudes [63,64,65]. Acoustic bubbles can not only be used to manipulate particles, cells, and other objects but also to manipulate fluid flows within microfluidic devices. For example, Ahmed et al. reported that using ultrasound to oscillate bubbles trapped in a “horse-shoe” structure inside microtubules could affect fluid flow (Figure 5a) [66]. Under the action of acoustic waves, the contact surface between the microbubbles and surrounding liquid vibrates, which triggers microstreaming [67,68]. In addition, medical microbubble contrast agents driven by ultrasonic pulse can be used for treatment delivery and monitoring [69].

In addition, Dijkink et al. reported that a semi-closed tube immersed in a liquid can be filled with a certain amount of gas. Under the action of acoustic waves, the gas is alternately discharged and pulled into the liquid through the open end of the tube (Figure 5b) [70]. The liquid leaves the pipe as a jet and enters the pipe simultaneously from the entire available stereo angle, forming a net momentum source that can drive this “acoustic scallop.” Therefore, the combination of acoustic bubbles with microtubules, microcavities, and various microstructures can generate forces for driving microrobots. In general, acoustic oscillation technology offers the selective excitation of bubbles, a wide range of action, simple equipment, easy implementation, and more.

### 2.5. Electrowetting-On-Dielectric (EWOD)

This section introduces the mechanism of the electrowetting method for bubble control. EWOD technology can change the contact angle between the droplets and dielectric layer; it can be used to manipulate the generation, transportation, mixing, and splitting of droplets [71,72]. When an electric voltage V is applied between the aqueous sessile droplet and electrode, the droplet spontaneously spreads out on the dielectric surface. The contact angle θ is modulated by the applied voltage according to the Lippmann–Young equation [73]:(10)$$\cos\theta =\cos\theta_{0}+\frac{\varepsilon \varepsilon_{0}}{2\gamma_{1g}t}V^{2},$$
where $\theta_{0}$ denotes the permittivity of the vacuum; ε is the dielectric constant of the dielectric layer; $\gamma_{1g}$ is the gas-to-liquid interfacial tension; t is the thickness of the dielectric layer; V is the electric potential; and $\theta_{0}$ is the initial contact angle at V=0 V.

Similar to droplet operation, Zhao et al. realized bubble transportation in any direction on a 2D surface using the EWOD principle [73]. The position of the bubble was controlled by the switching state of the electrodes underneath the hydrophobic dielectric layer. Activating the electrode on the left caused the bubble to move one step to the right. Likewise, the sequential activation of the subsequent electrodes produced continuous motion of the bubble. Subsequently, Chung et al. (of the same research group) combined the EWOD technology with acoustic oscillation to generate cavitation microstreaming around the bubble, and they realized the capture, transportation, and release of glass microbeads, fish eggs, and Daphnia via the moving bubble (Figure 6) [74,75]. EWOD and acoustic-based microbubbles are effective and noninvasive tools for processing micro-objects. The limitation of the EWOD method is that it can only be used to control the bubble’s position in the 2D plane. Therefore, it must be combined with other technologies (e.g., electrolytic water) to generate bubbles alongside the acoustic waves to excite the bubbles and strengthen the control thereof.

## 3. Bubbles Serving as Propulsion System

Bubbles can form an integral component in microrobots and provide their driving force. In this section, we review the micro/nanomotors that are propelled by bubbles generated via chemical reactions, and microswimmers propelled by bubbles driven by acoustic waves. Note that researchers in various fields refer to artificially prepared, cableless, movable, and controllable microstructures differently, using terms such as microrobots, micromotors, micromachines, and microswimmers; these were all classified as microrobots in this article.

### 3.1. Propulsion by Catalytic Reaction Generated Bubbles

In the past few decades, researchers have developed many different methods to drive and control micro/nanomotors. The propulsion mechanisms mainly include chemical propulsion (concentration-gradient propulsion, self-electrophoresis propulsion, and bubble-propelled), external-physical-fields driving (light field, magnetic field, and ultrasound), and hybrid propulsion [76,77,78]. Micro/nanomotors can be controlled by magnetic field, electric field, acoustic field, and other methods [79,80], and they have broad applications in biomedical [81] and water environments [82]. Among these, the micro/nanomotors that use catalytic reactions to produce bubbles and self-propel via the bubble recoil mechanism represent successful cases. This section describes the bubble-based propulsion of hollow tubular micromotors, Janus particles, and micromachines.

#### 3.1.1. Self-Propelled Tubular Micromotors

Tubular micromotors represent the most common self-propelled machines because of their simple structure, ease of manufacture, and multifunctionality [83]. Hollow tubular engines are typically manufactured by rolled-up nanotechnology [84] and template electrodeposition [85,86]; they consist of multiple layers of different metal curls. The research team led by Oliver first prepared hollow tubular nanomotors propelled by bubbles, which reduced the weight of the microrobot and improved its driving efficiency [87,88]. Bubble propulsion is the result of three phenomena: capillarity, bubble growth, and bubble extrusion [89,90,91,92]. In most tubular micromotors, the Pt layer acts as a catalyst to decompose the fuel in the solution (e.g., H_2_O_2_) and produce bubbles. The bubbles diffused and eventually broke or separated from the open end of the tubular micro/nanomotor. Once the bubble was ejected from one open end of the tube, the tube began to move in the opposite direction [93]. The Au interlayer was used to connect the Pt and Fe layers, whilst the outermost titanium (Ti) layer improved the mechanical strength of the micromotor [94]. The less expensive silver (Ag) catalyst can also be used for bubble propulsion [95]. Methods such as adding ferromagnetic layers of Fe, cobalt (Co), and nickel (Ni); optical control [96,97]; and ultrasound [98] can be used to control the direction or speed of the micromotor.

Bubble-based functionalized tubular micromotors are widely used in biomedicine [99,100]. For example, tubular micromotors loaded with biomolecules and particles can move to the required position and realize cargo collection and delivery. Wu et al. [96] modified the inner wall of a tubular micro/nanomotor using a thermally sensitive gelatin hydrogel layer containing Au nanoparticles and drugs. Near-infrared light produced considerable heat around the Au nanoparticles; this induced deformation of the hydrogel and led to rapid release of the drug. Zhang et al. [101] proposed a chemically powered jellyfish-like micromotor that could be propelled to a speed exceeding 209 μm/s via oxygen bubbles generated by catalase in 1.5% H_2_O_2_ fuel. Luo et al. [102] prepared a tubular micromotor containing an inner catalytic Pt layer, intermediate magnetic Ni layer, and outer cationic branched polyethyleneimine (PEI) layer. The micromotor was guided by a magnetic field and propelled by bubbles to extract nucleic acids efficiently. The fast movement of the bubble-propelled functional tubular micro/nanomotors means that they can also quickly sense and detect biomaterials. For example, a reduced graphene-oxide (rGO)/Pt double-layer tubular micromotor with a large number of active sites on the rGO surface can realize the rapid quantitative analysis of molecular concentration according to the fluorescence signal; this was used for the fluorescence detection of ricin [103]. In addition, functionalized tubular micromotors can also capture and collect molecules and particles whilst moving rapidly in solution, which can be applied to the fields of environmental purification, environmental monitoring, and pollutant degradation [104].

Clustering represents a future development direction for tubular micromotors [105]. Lu et al. [106] proposed dandelion-like microswarms assembled using catalytic tubular micromotors. Under ultrasonic excitation, the tubular manganese dioxide (MnO_2_) micromotor individuals were powered by the oxygen bubbles generated at their heads to swim rapidly in H_2_O_2_ solutions. When a large bubble core, generated via the fusion of multiple microbubbles, was excited and oscillated by ultrasound, these micromotor individuals gathered to it due to the locally intense acoustic field, to realize the dynamic assembly and cooperation of micromotors in a dandelion formation. Recently, Lu et al. [107] controlled the release and collection of tubular micromotor swarms. Hydrogen bubbles were produced at the tip of the charged electrode. When the bubbles oscillated driven by the acoustic field, they produced intensified vortexes, which spontaneously dispersed the tubular micromotors into the surrounding environment. By removing the attached bubbles, the sonoelectrode worked at a higher ultrasonic frequency to collect the micromotors on a large-scale via acoustic streaming (Figure 7a). Research into the precise control methods for individuals and clusters represents the inevitable development direction of bubble-propelled tubular micromotors. Meanwhile, developing non-toxic and biodegradable materials to replace the components and fuels of bubble-propelled tubular micromotors also represents a difficult problem that must be overcome in in vivo applications.

#### 3.1.2. Self-Propelled Janus Particles

Janus particles can be divided into three categories according to their structures and dimensions. These three categories are one-dimensional (1D) cylinders, 2D disc-shaped particles, and 3D spherical Janus particles. The lack of central symmetry is an inherent feature shared by these particles [108,109,110]. In 2004, two research teams from Pennsylvania State [111,112] and the University of Toronto [113] independently proposed that bimetallic nanorods be used as nanocatalytic motors in H_2_O_2_ solutions. At one end of the micromotor, self-propulsion was realized by catalyzing the decomposition of the H_2_O_2_ solution to produce oxygen bubbles. This cylindrical micromotor was only a few hundred nanometers in diameter, making it a real nanomotor. Micromotors propelled by bubbles generated by the asymmetric position can overcome the limitations of the “scallop theorem” and realize non-cyclic movement on a small-scale. Therefore, this driving method has attracted the attention of researchers, and self-propelled microrobots of various shapes have emerged. Howse, Gibbs, and Walther successively prepared asymmetric and flexible Janus spheres [114,115,116]. When the diameter of the Janus microspheres was less than 5–10 μm, they were self-propelled by the self-diffusion phoresis mechanism. When the diameter was 10 μm above the surface, the oxygen produced via decomposition nucleated and gathered to form bubbles. In this case, the self-driving motion was realized by the microbubbles [117]. H_2_O_2_ is currently the most popular fuel for chemical propulsion [118,119]. Au- [120,121], Pt- [122,123], Fe- [124,125], Ag- [126], Mg- [127], and other metal-based micro/nanorobots can also decompose H_2_O_2_ to generate bubbles. The rebound force generated by the bubbles can promote the movement of micro/nanorobots. The speed of the catalytic reaction can be controlled by changing the concentration of H_2_O_2_ to adjust the speed of the nanomotor. The moving direction can be controlled using a magnetic field or other methods [128,129], and drug-triggered delivery can be realized by combining it with a light field [130,131].

Due to their efficient propulsion, cargo traction, accurate motion control, and design versatility, these chemically powered micromotors have been proven capable of performing various biomedical tasks. Diez et al. [132] designed an innovative multifunctional gated Pt-mesoporous silica nanomotor, which used Pt as the propulsion element, mesoporous silica nanoparticles as the drug-loading element, and a disulfide-containing oligo (ethylene glycol) chain (S−S−PEG) as a gating system. Under bubble propulsion via the catalytic reduction of H_2_O_2_, it could move directionally and release drugs to kill cancer cells. Micro/nanomotors driven by hydrogen bubbles can perform medical tasks in acidic environments (e.g., the stomach) using the chemical reaction of Mg or Zn in acid [133,134]. Karshalev et al. [135] prepared a microrobot with Mg/TiO_2_ as the core; this could be propelled in a gastric acid environment and could transport Fe and selenium (Se) to treat iron deficiency anemia. Lin et al. [136] prepared a bubble-propelled Janus gallium (GA)/Zn microrobot. This microrobot can be propelled at a speed of 383 µm/s in a simulated gastric acid (pH = 0.5) environment via hydrogen bubbles generated by the Zn-acid reaction. It offers good biocompatibility and biodegradability and can be used for the targeted treatment of gastrointestinal bacteria such as H. pylori (Figure 7b). However, hydrogen peroxide solution is harmful to the human body. The micro/nanomotors based on this chemical fuel have poor biocompatibility and cannot be truly applied to the biological environment. Undoubtedly, bubble micromotors driven by chemical reactions in the human physiological environment have high biocompatibility and good development potential. For example, the micromotor based on the Mg/Zn-acid reaction is more likely to work in the human gastrointestinal tract.

#### 3.1.3. Self-Propelled Micromachines

In 2002, the research team led by Whitesides manufactured a motor driven by an asymmetric chemical reaction [137]. We describe this groundbreaking work in this section because it is irregularly shaped and bears a closer resemblance to a miniature machine. The self-driving machine consisted of a thin plate made of polydimethylsiloxane (PDMS) and a porous glass filter connected by a stainless-steel pin. Pt-coated glass can catalyze the decomposition of H_2_O_2_ to produce bubbles, which can be used as a motor to drive the PDMS plate. Zhu et al. [138] fabricated free-swimming microfish composed of poly(ethylene glycol) diacrylate (PEGDA)-based hydrogels and functional nanoparticles using a rapid 3D printing technology called microscale continuous optical printing (µCOP). The catalytic Pt nanoparticles were polymerized in the tail of the microfish so that the fish could achieve self-propulsion by decomposing the H_2_O_2_ solution; this achieved a speed of 780 µm s^−1^ in a 15% peroxide solution. Magnetic Fe_3_O_4_ nanoparticles were polymerized at the head of the microfish, which facilitated alignment and guidance of the fish using a remote magnet. In addition, other functional toxin-neutralizing nanoparticles could be incorporated into the hydrogel matrix of the fish body to explore their detoxification applications. The motion of a structure composed of multiple parts (connected by joints) can imitate the motion of a biological system. In 2017, Yoshizumi et al. [139] connected two Au/Pt micromotors using a polymer tube as a joint formed by stacking cationic poly(allylamine hydrochloride) (PAH) and anionic poly (acrylic acid) (PAA) by layer-by-layer technology. The stiffness of the polymer joint could be accurately controlled by adjusting the thickness of the polymer layer, and the equilibrium bending angle between the two motors could be adjusted by heat or chemical treatment. They realized bending and rotation of the Pt/Au–Au/Pt and Pt/Au–Pt/Au structures, respectively.

Recently, Li et al. [140] used a two-photon absorption-based direct laser writing technique to prepare a fish-shaped microrobot with a serrated tail and used sputtering deposition technology to coat a Ni layer with a thickness of 200 nm and a Pt layer with a thickness of 100 nm onto the vertical structure on the substrate. The Ni and Pt layers were responsible for the external magnetic response and chemical catalysis, respectively. Therefore, the microrobot could achieve high-speed motion and magnetic steering control in an H_2_O_2_ solution. More reaction channels mean that the number of propulsion bubbles and the speed of the microrobot increase. Therefore, the multi-channel and nano interface on the serrated tail can considerably increase the catalytic reaction and allow the microrobot to move at a high speed (Figure 7c). These micromachines have unique structures, which expand our range of possibilities and may inspire researchers to design microrobots with diverse architectures and efficient motion performances to explore the further applications of bubble-propelled microrobots based on catalytic reactions. Furthermore, intelligent micromotors integrating induction, judgments, and responses have become new development trends.

**Figure 7 micromachines-13-01068-f007:**
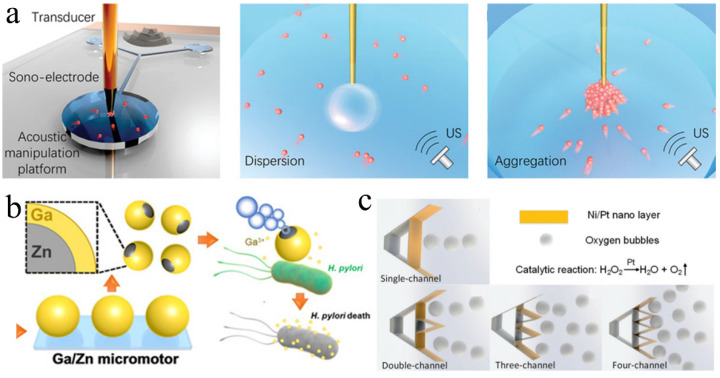
Chemically driven microrobots. (**a**) Acoustic manipulation platform and sonoelectrode-enabled dual swarming modes including dispersion and aggregation. Adapted from Lu et al. [107] with permission from John Wiley and Sons, Copyright 2021. (**b**) Fabrication and antibacterial process of Janus Ga/Zn micromotors. Adapted from Lin et al. [136] with permission from John Wiley and Sons, Copyright 2020. (**c**) Chemical swimming microrobots with serrated-tail-enhanced propulsion interfaces by O_2_ bubble production during H_2_O_2_ decomposition. Adapted from Li et al. [140] with permission from the RSC Pub, Copyright 2018.

Although bubbles are not a part of these microrobots, they provide power to them. In another bubble driving method, the bubble is a component of the microrobot body and generates a driving force through vibration of the gas–liquid interface under the excitation of an acoustic field, as presented in Section 3.2.

### 3.2. Propulsion by Acoustic Oscillating Bubbles

Microswimmers have considerable potential for various biomedical applications including targeted drug delivery [141], microsurgery [142,143], and medical innovations [144]. When considering the propulsion and control of microswimmers in actual biomedical environments, various propulsion principles have been studied including biofuel [145], chemical fuel [146], and magnetic drive [147]. However, their practical applications are limited. Acoustic actuation is promising and attractive because it is noninvasive and biocompatible [148,149]. In an acoustic bubble-driven microrobot, the bubble itself, as an “engine,” generates a driving force under the action of an acoustic field and becomes an indispensable part of the robot body [150]. The speed of microswimmers can be easily controlled using acoustic waves with different ultrasonic frequencies. In 2011, Won et al. first prepared a micron/milli-sized open box using an Al film and verified that when the acoustic wave propagated to the bubbles on the surface of the box in the liquid medium, those bubbles oscillated and generated cavitation microstreaming, which could be used to promote small floating objects [151]. In 2015, Ahmed et al. [152] demonstrated the use of oscillating bubbles to drive microswimmers. They used micro electro mechanical system (MEMS) technology to create a microstructure with a semi-closed pipe. The bubbles in the microchannel were excited by the acoustic field to generate thrust at the gas–liquid interface vibration. The microrobot could realize straight-line, turning, and other motions by selectively exciting bubbles in different positions. Subsequently, they prepared a type of soft microswimmer polymerized with a superparamagnetic particle chain, which could move under the action of acoustic bubbles; they controlled its direction using a magnetic field [153] (Figure 8a). The microrobot could push and pull particles and cells. Compared with chemical driving, microrobots driven by acoustic bubbles require no fuel and reduce the requirements of the liquid environment, which makes them more suitable for use in organisms [154,155].

Subsequently, an increasing number of researchers have designed a variety of microswimmers. Feng et al. [156] designed and fabricated a 1D parylene microchannel with a single end opening using microphotolithography technology. The bubbles trapped in the microchannel were oscillated by the external acoustic field, and they periodically generated an inlet/exhaust microstreaming flow through the opening of the microchannel. Under an increase in the Reynolds number, the difference between the intake and exhaust flow increased and finally produced a net flow to drive the microchannel devices. Subsequently, they designed a 2D microswimmer with bi-directional (linear and rotational) propulsion steering based upon acoustic oscillation bubbles alone [157]. To realize 3D movement of the microswimmer, they combined microtubules of different lengths and used acoustic waves of different directions and frequencies to excite the bubbles. However, the size of the microrobot body increased significantly [158] (Figure 8b).

To prolong the service lives of microbubbles, Bertin et al. [159] prepared armored microbubbles (AMB), which ensured that the microbubbles could produce contactless acoustic propulsive flows for several hours under stable conditions. Subsequently, they produced double and triple propulsors based on armored bubbles [160], and more recently studied the multi-directional streaming flows generated by AMBs with multiple surface holes under ultrasound excitation [161]. Louf et al. designed a hovercraft-mimicking microswimmer with only one bubble, which could hover on a base and move [162]. Ren et al. [33] designed a microswimmer based on a bubble half-capsule shape, which was controlled by acoustic and magnetic fields. The secondary Bjerknes force and locally generated acoustic streaming propulsive force affected the microswimmer. The combined force of the two forces enabled it to swim independently under the guidance of a magnetic field in a 3D space. By changing the shape of the microrobot and introducing the design of “fin”, the movement direction of the microrobot could be controlled, and climbing in the pipeline could be realized (Figure 8c) [163]. Subsequently, Aghakhani et al. manufactured an acoustic bubble capsule microrobot that could be propelled at a high shear rate. They demonstrated the effective propulsion of this microrobot in various biological fluids and conducted in vitro navigation through the mucus layer on the biological 3D surface [164]. Recently, Luo et al. [34] designed a microswimmer based on two different bubbles, which could be completely promoted and manipulated via an ultrasonic transducer and exhibited boundary-following characteristics similar to biological swimmers; this may improve the current technologies for targeted drug delivery (Figure 8d).

In addition, rotational micro-propellers can be manufactured using the driving force of acoustically actuated bubbles. In 2018, Jang et al. proposed a novel acoustic energy harvesting technique using periodically vibrating piezoelectric cantilever beams (driven by synthetic jets induced by acoustic oscillation bubbles) to generate electrical energy [165]. Subsequently, Dincel et al. developed an acoustic frequency driven microbubble motor (AFMO) device and achieved high-speed rotation [166]. Recently, Mohanty [167] manufactured a magneto-acoustically actuated micro-propeller, which could be used not only as a mobile microfluidic mixer, but also as an automatic propulsion microrobot for directional operation (Figure 8e).

In recent years, bionic bubble microrobots have become a popular research topic. Mohanty et al. [168] fabricated an unconstrained microrobot called CeFlowBots, which contained a set of acoustic resonance bubbles that pumped liquid through its body to improve propulsion. The CeFlowBots were manipulated under the combined influence of magnetic and acoustic fields to grasp and release objects in the workspace. CeFlowBots can be navigated in a remote environment and can perform directional operations for drug delivery (Figure 8f). Acoustic bubble microrobots with smaller sizes, enhanced motion capabilities, improved control performance, and wider application ranges represent the future development directions. For microrobots driven by multiple bubbles of different sizes, coupling resonance may occur between bubbles under excitation of the acoustic field; this usually prevents the microrobots from being accurately located. Therefore, the combination of acoustic and magnetic fields is necessary, particularly for microrobots applied inside the human body.

**Figure 8 micromachines-13-01068-f008:**
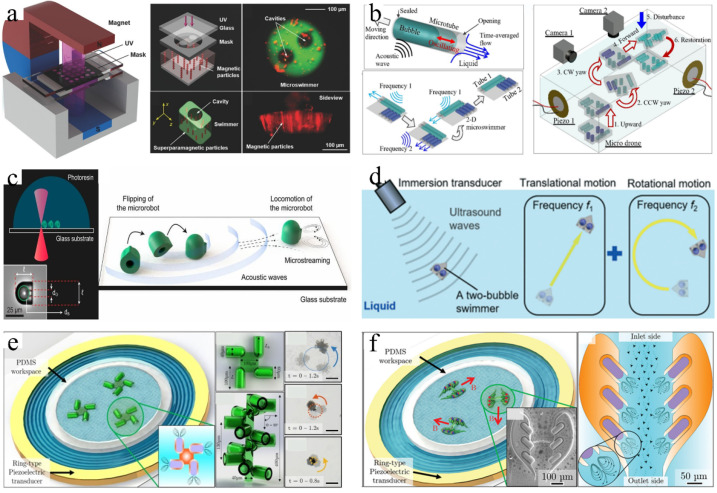
(**a**) A schematic of the acousto-magnetic soft microswimmers fabrication setup, the magnetic particles and cavity inside the microswimmer. Adapted from Ahmed et al. [153] with permission from John Wiley and Sons, Copyright 2017. (**b**) 1D microtube, 2D microswimmer, and 3D microdrone powered by acoustic microbubbles. Adapted from Liu et al. [158] with permission from The Royal Society of Chemistry, Copyright 2021. (**c**) Fabrication of the microrobot and schematics of the microrobot propulsion. Adapted from Aghakhani et al. [163] with permission under the terms of CC BY-NC-ND 4.0 License, Copyright 2020. (**d**) Microswimmer powered and steered by an ultrasound transducer in a fluid environment as well as its swimming trajectory. Adapted from Luo et al. [34] with permission from Royal Society of Chemistry, Copyright 2021. (**e**) Acoustic actuation test-bed, two types of propellers, and time-lapse images of the propeller. Adapted from Mohanty et al. [167] with permission under the terms of the CC BY License, Copyright 2021. (**f**) Streaming pattern and resultant flow of the magneto-acoustic-actuated CeFlowBots. Adapted from Mohanty et al. [168] with permission under the terms of the CC BY License, Copyright 2021.

## 4. Bubbles Serving as Micromanipulators

Bubbles may not form the main body of microrobots but do play an irreplaceable manipulation role. In this section, we review the bubbles that serve as operators in combination with 3D motion machines such as micropipettes and microrobots, and bubbles that serve as generators and transmission components to realize energy conversion.

### 4.1. Bubble Operators

Bubbles can adsorb and release micro-objects under the excitation of an acoustic field; thus, it can function as a manipulator for microrobots. Bubbles can be attached to the surface of the microrobot to perform operations. Chung et al. placed an acoustically excited oscillating bubble at a hydrophobic micro-rod top to realize the adsorption and movement of particles in a 3D space [169]. Lee et al. [170] generated bubbles on a microfabricated chip composed of tip-shaped electrodes by controlling the applied voltage and time of electrochemical methods. Subsequently, two bubbles were successively transferred to the tip of the U-shaped rod coated with a hydrophobic layer, and the fish eggs and glass beads were manipulated under the excitation of the acoustic field. In 2016, Ahmed et al. realized the rotation of particles based on the torque induced by the hydrodynamic flow field generated by the acoustic bubble microstreaming. Inspired by this, the bubble operators further realized 3D rotation and graph reconstruction of the particles and cells. Recently, Li et al. contained a microbubble at the top of a micropipette, and generated secondary radiation force and microstreaming to capture and rotate microbeads in an aqueous medium via acoustic vibrating microbubble near its resonant frequency [171]. Using the same principle, Zhou et al. [172] generated a bubble at the tip of the micropipette of a 3D gripper, and the bubble volume was controlled by adjusting the pressure in the micropipette. By replacing the micropipette with different needle tip sizes, bubbles with diameters of 20 µm to 1 mm were obtained. Due to the secondary acoustic radiation force, the neighboring microparticles were attracted to the gas–liquid interface and transported to the desired position by the micropipette. When the transducer was turned off, the objects were released (Figure 9a).

For a microrobot moved and steered by a magnetic field, the bubble can be attached to its surface or contained in its interior. Kwon et al. realized the wireless control of bubbles using a magnetic control microrobot instead of a micro-rod [173]. They used the injection method to adsorb a bubble at the bottom of the magnetic plate, and the magnetic field generated by three pairs of electric coils could wirelessly control the robot and bubble in 3D space. The acoustic oscillating bubble could grasp and move the microbeads or cells to a designated position; then, the objects could be released by turning off the ultrasound. Park et al. [174,175] proposed a microrobot that primarily consisted of a compressible bubble (that functioned as a gripping tool) and a pair of permanent magnets. Using an external magnetic controller to control the 2D motion of the microrobot, the acoustic oscillation bubble on the microrobot could carry out cell enucleation. Giltinan et al. [12] designed a magnetic microgripper using the bubble capillary force. The microgripper was composed of a cuboid containing at least one cavity and the bubbles thereby captured. The bubbles bore the capillary force and maintained the grip of the parts during movement of the microgripper, which was controlled by a magnetic field. When the microrobot sizes are further reduced to the millimeter level, they are expected to be able to enter blood vessels or other biological pipelines to complete specific tasks [176,177]. Kwon et al. further realized micro-object manipulation in a microfabricated channel using an electromagnetically driven microrobot with an acoustically oscillating bubble. The acoustic oscillating bubble allowed the microrobot to realize more flexible, accurate, noninvasive, and harmless micromanipulation, which has great application value in the analysis of living cells and biological samples.

### 4.2. Bubble Generator and Transmission Components

Bubbles actuated by acoustic waves can be used as a source of mechanical energy for “generators.” Jeon et al. [178] suspended a bubble on a flexible piezocantilever. When the bubble was excited by acoustic waves near its resonance frequency, it oscillated and generated cavitation microstreaming around it. The microstreaming bent the piezocantilever with fine vibrations, which caused the piezocantilever to generate electric power (Figure 9b). The generated voltage mainly depended on the frequency of the applied acoustic field and the size of the bubble, and it was inversely proportional to the distance between the bubble and piezoelectric actuator [179]. They also discovered that the electrical output could be improved by increasing the number of bubbles.

Using the light-induced thermal convection and the bubble’s surface tension, bubbles can be used as “transmission parts” in microrobots or microcomponents to provide the required force. Villangca et al. [13] proposed an optical driven micro-tool to realize a syringe function. They fabricated the “chassis” and “wheels” of the micromachine via two-photon-polymerization and deposited metal layers inside the micromachine via electron beam physical vapor deposition. The motion of the machine was manipulated using the four captured beams. When light shone on the inner metal layer, a bubble was generated, and the resulting thermal convection could be used to load and unload cargo (Figure 9c). In addition, Jiang et al. realized the reversible connection between SU-8 components using a directional jet and the surface tension of bubbles generated via a laser [180].

**Figure 9 micromachines-13-01068-f009:**
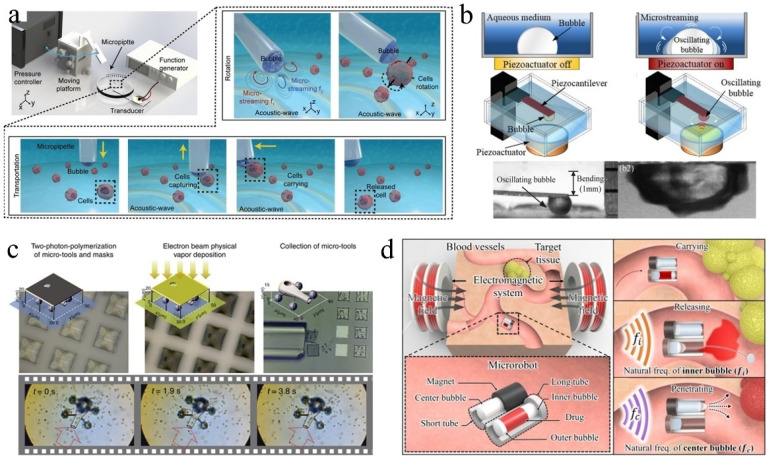
(**a**) The rotation and transportation motion of an acoustic bubble-based noninvasive microgripper. Adapted from Zhou et al. [172] with permission from John Wiley and Sons, Copyright 2021. (**b**) Upper: energy harvesting mechanism based on an acoustically oscillating bubble. Lower: still images of the oscillating bubble. Adapted from Jeon et al. [178] with permission from Springer Nature, Copyright 2021. (**c**) Upper: fabrication of micro-tools for material transport. Lower: collection of polystyrene beads next to the micro-tool used as a tiny pump. Adapted from Villangca et al. [13] with permission under the terms of the CC BY-NC-ND 4.0 License, Copyright 2016. (**d**) Acoustic bubble-based drug delivery technology: drug carrying, releasing, and penetrating. Adapted from Jeong et al. [181] with permission from Elsevier, Copyright 2020.

Bubbles can also act as the “controllable switch” of microrobots to undertake the tasks of drug delivery, drug release, and drug penetration promotion in the body (Figure 9d) [181]. The drug was stored in a long tube of a double tubular microrobot, sealed between two bubbles of different volumes. The most giant bubble was placed inside the short tube, and the tail was a magnetic component used for driving and navigation. Driven by acoustic waves of different frequencies, the inner bubble in the long cavity released the outer bubble and drugs through oscillation; then, the largest bubble in the short tube oscillated to accelerate the diffusion of drugs. In these microrobots, bubbles represent the medium of energy conversion including the conversion of acoustic, light, and heat energy into kinetic energy; hence, these microrobots can complete a variety of functions.

## 5. Bubbles Serving as Microrobots

Bubbles can be used as microrobots to manipulate and assemble particles, cells, and microstructures. In this section, we review the bubble microrobots that can perform dexterous and complex microassembly tasks in 2D and 3D spaces.

### 5.1. Bubble Microrobots Used in 2D Manipulation and Assembly

Whether driven by EWOD or optically, bubbles can move across the surface of the chip. Moreover, both methods can be combined with an acoustic field to vibrate the bubble and realize the grasping and movement of micro-objects, although these operations are only on the 2D scale. This section describes the EWOD-driven, optothermal, and bubble microrobots combining light and acoustic fields.

#### 5.1.1. EWOD-Driven Bubble Microrobots

Researchers at the University of Pittsburgh have applied EWOD technology to manipulate bubbles. When the electrodes are energized in turn, bubbles can move on the electrode surface (Figure 10a) [182], and the mixing and splitting of bubbles can also be realized [73]. Alternating current-EWOD (AC-EWOD)-driven bubbles can be used to enhance fluid mixing in microfluidic chips and perform micromanipulation and particle separation [183,184]. In 2012, Lee et al. used bubbles in EWOD to promote the mixing of microfluidics [184]. Subsequently, they realized the driving of two bubbles in the EWOD [185]. To improve the controllability of the manipulation, double bubbles were used to form a flow field with a fixed direction; this could be used to realize the directional transport of fish eggs. Recently, Yan et al. induced the escape, coalescence, and departure of bubbles activated by an electric field by using a simple EWOD device [186]. When the copper electrode wire (inserted into deionized water) was placed on one side of the bubbles, the bubbles escaped from their initial positions with the opening of the voltage. The pin-on-one-side EWOD device could continuously push bubbles out of their initial position under a low-frequency voltage, without the need for a complex patterned electrode array channel design. Sun et al. [187] observed standing waves on a millimeter-sized bubble surface on the substrate of an EWOD device and discovered that the amplitudes of these waves varied with frequency, which could identify three formants. However, the limitation of EWOD technology in operating bubbles is that, although it can control the positions of bubbles, MEMS chips with electrodes must be designed and manufactured in advance, and the route of bubble movement is limited by the arrangement of electrodes and is restricted to the 2D plane. In addition, bubbles must be excited by acoustic waves to have a strong adsorption capacity for manipulating micro-objects.

#### 5.1.2. Optothermal Bubble Microrobots

Compared to the EWOD method, the bubbles generated and controlled by the optothermal method can move freely on the chip under movement of the laser spot and do not require ultrasonic excitation to manipulate micro-objects. In 2007, Ohta et al. first demonstrated the movement of optothermal bubbles formed via laser absorption on an amorphous silicon substrate [14]. Hu et al. [51] showed that optothermally generated bubbles could move and arrange multiple triangular SU-8 photoresist microstructures. Subsequently, they used a laser-induced bubble to drive a disc-shaped PEGDA hydrogel microrobot with a conical groove at the bottom [54]. One or a pair of cooperative bubble microrobots could assemble 20 μm-diameter polystyrene beads, operate individual yeast cells, and assemble a square agarose microgel encapsulated with cells into a 3 × 4 array. They also designed a T-shaped bubble-driven light-absorbing microrobot [188]. In 2014, Zhao proposed that under the synergistic action of Marangoni convection and the surface tension of an optothermal bubble microrobot, an adsorption force could be applied to the particles suspended in a solution. When the bubble moves on the solid–liquid interface, it can pull the microparticles to move with it (Figure 10b) [44].

In terms of cell operation, optothermal bubble microrobots can realize cell movement and deformation [189], perforation, and lysis [190]. In 2013, Hu et al. [55] proposed an opto-thermocapillary micromanipulator (OTMm). Intermittent laser irradiation was used to induce a bubble to quickly separate from the substrate and float upward, forming a light thermocapillary convection around it to move the cells at a certain distance. When a fixed-frequency laser was used to irradiate the chip, an endless stream of bubble microrobots could drive the cells to move continuously. This non-contact operation method caused almost no damage to the cells. Later, Fan et al. of the same research group used the oscillation of microbubble size (caused by the opening and closing cycle of a laser pulse) to generate shear stress on the nearby cell membrane to realize cell perforation [56]. The perforation efficiency was further improved by generating microbubbles with size oscillations directly below the target cell [191]. Subsequently, they integrated a single-cell analysis platform for cell perforation, lysis, and manipulation (Figure 10c) [190]. Yuan et al. [192] studied the cell membrane deformation and biological effects caused by the jet flow generated by laser-induced tandem bubbles (TBS) at the single-cell level. In addition, Fujii et al. [193] demonstrated that optothermal bubbles could manipulate a single DNA nanowire and cross-wire formation, which means that optothermal bubbles can manipulate other biological macromolecules. Visible-light thermal bubble microrobots have broad application prospects in biomedicine and assembly engineering [194,195]. Multiple bubble microrobots can have a synergistic effect [196,197]. Rahman et al. generated and manipulated multiple opto-thermocapillary flow-addressed bubble (OFB) microrobots at the same time by using the spatial light modulator (SLM) to realize independent and cooperative operation tasks (Figure 10d) [198,199]. They realized the controllable movement of micromodules and glass beads and improved the operation efficiency of bubble microrobots.

#### 5.1.3. Bubble Microrobots Combining Light and Acoustic Fields

The high temperature on the surfaces of optothermal bubbles has benefits and drawbacks. It is conducive to cell perforation and lysis but causes severe damage when directly operating cells. To avoid possible thermal damage and improve the biocompatibility of optothermal bubbles, Xie et al. proposed optoacoustic tweeters [200]. They used the optothermal effect to generate bubbles of appropriate sizes at the designated position of the chip; then, they turned off the laser and used acoustic waves to induce the surface bubbles to oscillate at the resonant frequency. Thus, the particles/cells were captured locally around each bubble. Subsequently, they used acoustic microstreaming around the oscillating bubble to measure the deformation ability of the cell (Figure 10e) [48]. In addition to the non-destructive operation of cells, researchers have discovered that bubble microrobots combined with light and acoustic fields also have other characteristics including the simultaneous collection or dispersion of batch objects as well as the classification of particles of different sizes. In 2017, Dai et al. [201] proposed a method for the simultaneous operation and transmission of multitarget objects in an unclosed chip. The laser was turned off and the piezoelectric actuator turned on at a certain position, and the objects gathered to the vibrating laser-generated bubble. When the laser was turned on again, the increasing volume and broken bubbles produced dispersion of the collected small objects. The above steps were repeated at another location to realize the transfer of the object groups. Shin et al. [202] observed that when a bubble oscillated around its resonance frequency under acoustic excitation, it attracted large particles and repelled small particles due to the competition between the secondary acoustic radiation force and drag force to classify the particles. We anticipate more applications of microrobots based on light and acoustic fields in particle and microstructure operations.

### 5.2. Bubble Microrobots Used in 3D Manipulation and Assembly

Whether using EWOD technology or laser to drive and control bubbles, bubble microrobots can only be used for the 2D operation and assembly of microparticles, cells, or microstructures. When microbubbles are blown into the liquid via an air injector, the microflow floats the micromodule and contributes to the 3D assembly of the hydrogel modules. Fukuda et al. used the flow field generated by bubble motions to perform a 3D operation upon cell-laden hydrogel micromodules to manufacture biological tissue in vitro [203,204]. An air injector was used to blow microbubbles into the solution to make the prepared micromodule float up. Simultaneously, another holding pipette was used to collect the micromodules. When the holding pipette connected the blown micromodules in series, the posture of each module had to be adjusted to align it. The flow caused by the bubble movement can also play a role in the arrangement of micromodules (Figure 10f) [205]. Although the micromodules can realize 3D movement under the action of the flow field, the controllability of the bubbles and micromodules is limited. This micromodule operation and assembly method requires two actuators to control the syringe and collector. Moreover, the technology places certain requirements upon the shape of the micromodules, and there must be holes in the center to allow it to be picked up, collected, and aligned.

To solve these problems, Dai et al. used a 2D optothermal bubble microrobot to realize the 3D operation and assembly of micromodules, which provided more opportunities for the construction of biological tissues in vitro (Figure 10g) [43]. When a bubble microrobot was generated at the bottom of the micromodule, the micromodule could be lifted and turned over. The 3D gesture of the micromodule could be adjusted by changing the generation position and movement direction of the bubble. For example, the circular structure of a hydrogel module could be assembled into a 3D tubular structure with the shape characteristics of vascular tissue. Two annular modules of different sizes could also be nested together to provide a possible method for analyzing the roles of cells at different positions in blood vessels as well as the cross-vascular transmission of substances. On this basis, they used the 3D operation abilities of microbubbles to assemble micromodules with different interfaces into an interconnected whole, referred to as an integrated assembly (Figure 10h) [206]. Multiple bubble microrobots cooperated to lift, put down, fix, and move the micromodule to thereby realize the assembly of tenon and mortise, gear, chain, car, and various other microstructures. Integrated assembly technology based on microbubbles provides a new solution for the manufacturing, assembly, and development of micro/nanostructures. Recently, Ge et al. used these bubble microrobots to manipulate cell-loaded microstructures fabricated using a digital micromirror device (DMD)-based optical projection lithography system to form peritoneal tissue, which was similar to the biological peritoneum in terms of the mechanical properties, surface morphology, and internal microstructure [207]. The 3D operation and assembly method proposed by Dai et al. have excellent application prospects for in vitro biological tissue construction, microassembly factories, and more.

**Figure 10 micromachines-13-01068-f010:**
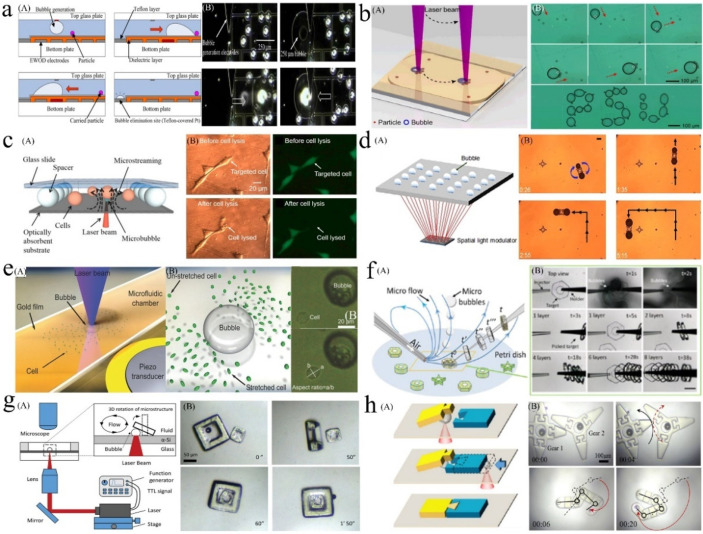
(**a**) A: Schematic of a bubble-based manipulator with three operations: creation, transportation, and elimination. B: Manipulation of micron-sized particles using the bubble transported by EWOD. Adapted from Chung et al. [182] with permission from the Institute of Physics Publishing, Copyright 2008. (**b**) A: Illustration of the particle manipulation process. B: Collecting and manipulating randomly distributed polystyrene particles to trace the letters “P”, “S”, and “U”. Adapted from Zhao et al. [50] with permission from The Royal Society of Chemistry, Copyright 2014. (**c**) A: The 3D structure of the microfluidic chamber to culture and lyse cells. B: Brightfield and fluorescent images before and after the adherent single-cell lysis. Adapted from Fan et al. [190] with permission under the terms of the CC BY 4.0 License, Copyright 2017. (**d**) A: A single laser beam shaped by a spatial light modulator to create dynamic optical patterns that facilitate the simultaneous control of multiple OFB microrobots. B: Manipulation of a microstructure using a pair of OFB microrobots. Adapted from Rahman et al. [199] with permission under the terms of the CC BY 4.0 License, Copyright 2017. (**e**) A: Schematic for the configuration of a microfluidic chamber and piezo transducer. B: Schematic for cell stretching around an acoustically activated oscillating bubble as well as images of a stretched spherical-shaped suspended cell. Adapted from Xie et al. [48] with permission from John Wiley and Sons, Copyright 2016. (**f**) A: Schematic of the sequential pickup of micromodules based on microbubble injection. B: Pickup of cell-laden micromodules with the hexagonal structure from the solution. Adapted from Wang et al. [205] with permission from The American Chemical Society, Copyright 2017. (**g**) A: Schematic of the system setup used to achieve the 3D manipulation of the microstructures via optothermal bubbles. B: The 3D manipulation and nested assembly of square ring microstructures. Adapted from Dai et al. [43] with permission from John Wiley and Sons, Copyright 2019. (**h**) A: Schematic of the integrated assembly process of two microparts using multifunctional optothermal bubble microrobots. B: The transmission of two gears and movement of the snake-shaped structure after assembly. Adapted from Dai et al. [206] with permission from The American Chemical Society, Copyright 2020.

## 6. Summary and Outlook

Micro/nanobubbles have become a general and diversified tool preferred by an increasing number of researchers as they have broad research prospects. Microrobot technology, in which bubbles play an irreplaceable and pivotal role, has developed rapidly. This paper reviewed the recent research results for bubbles combined with microrobots. First, the methods for generating and controlling bubbles were introduced; these are an important theoretical basis and prerequisite for manufacturing bubble-based microrobots. Then, the three functions of bubbles in the microrobot field—propulsion, manipulation, and assembly—were introduced. Based on their roles in the field of microrobots, bubbles were divided into bubble-driven microrobots, bubble operators combined with microrobots, and bubble microrobots. The mechanisms related to these three roles are listed in Table 1. In addition, the bubbles’ production methods and service lives as well as the advantages, limits, and applications of each mechanism, were summarized.

Microbubbles have opened up a novel path for the development of microrobot technology and are playing a more important role in microrobotics; however, bubble-based microrobots still suffer from some problems including short bubble service lives, reduced controllability of bubbles (because of time offsets), simple functions, lack of intelligence, and so on. To solve these problems, several simple methods have been proposed such as applying chemical hydrophobic surface modification [163] or designing special cavities of microstructures [164] to enhance the stability of bubbles and prolong their service lives, combining multiple fields to expand the functions of microrobots, and introducing control algorithms to improve their intelligence. The challenges and possible directions for the future development of bubbles in the field of micro/nanorobots are discussed in detail below.

Multi-field combination of bubble microrobots

Aside from the optothermal effect, it is difficult to drive and control a bubble microrobot relying only upon a single physical field. For example, microswimmers must rely on bubbles excited by an acoustic field (to provide a driving force) and magnetic field (to control the motion trajectory) [34]. Recently, the combination of bubble propulsion based on chemical reactions with acoustic [107] or magnetic fields [140] has become a research hotspot. The combination of multiple physical fields not only satisfies more complex and multifunctional applications, but also solves the problems and limitations of a single physical field. For example, to avoid the influence of high temperature in the laser area of an optothermal bubble microrobot on cells, non-contact operation was realized by combining it with an acoustic field [200]. The compatibility of multiple physical fields may form the basis for the integrated functionalization of bubble microrobots in the future. We hope that bubble microrobots can achieve richer capabilities and perform more comprehensive functions through the combination of multiple fields.

**Table 1 micromachines-13-01068-t001:** The role of bubbles in the field of microrobots.

Role/Mechanism	Production	Service Life	Advantage	Limit	Application	Ref.
Propulsion/chemical reaction generated bubbles	Chemical reaction	Short, bubbles are generated and quickly separated from the microrobot	Fast driving speed	The generation of bubbles and the movement performance of microrobot is affected by the consumption of chemical fuel; low biocompatibility	Biomedicine, biological detection, environmental purification	[107,136,140]
Propulsion/acoustically excited bubbles	Direct acquisition	Related to the shape and hydrophobic properties of the bubble-containing structures and acoustic excitation parameters	Simple and biocompatible equipment	Under long-time acoustic excitation, the change of bubble volume leads to a change of resonance frequency, which affects the motion performance of the microrobots	Targeted drug delivery, microsurgery, manipulation	[34,153,158,163,167,168]
Manipulator/manipulator based on acoustic bubble	Direct acquisition	Related to the acoustic excitation parameters	Adsorbable, removable, noninvasive, and flexible	Combined with manipulator, acoustic field, or magnetic field; complex structure	Analysis of living cells and biological samples, manipulation	[172]
Manipulator/bubble engine and transmission component	Direct acquisition, optothermal effect	For acoustic bubbles, service life relates to the structural design and acoustic excitation parameters. For optothermal bubbles, it depends on the opening and closing of the laser	Energy conversion, simple structure, and strong controllability	Weak change of flow field caused by bubble generation, oscillation or rupture; limited energy conversion	Energy conversion, cargo transportation, drug delivery	[13,178,181]
Microrobot/EWOD technology (2D)	Direct acquisition, chemical reaction	Related to the size of generated bubbles and acoustic excitation parameters	Controllable movement and low energy consumption	Limited movement of bubbles due to the electrode arrangement on the chip	Fluid mixing, micro-object manipulation	[182]
Microrobot/combining light field and acoustic field (2D)	Optothermal effect	Related to the opening and closing of the laser and acoustic excitation parameters	Improves the biocompatibility of optothermal bubbles	Narrow application range	Manipulation, particle classification	[48]
Microrobot/optothermal effect (2D/3D)	Optothermal effect	Depends on the opening and closing of the laser	Controllable and flexible bubble position and volume	Limited biomedical applications because of the high temperature around bubbles	Fluid control, cell lysis, manipulation, and assembly	[43,50,190,199,206]

Research and application of bionic microrobots based on bubbles

In recent years, functional bionic robots have become a focus of scientific research. When a jellyfish swims in seawater, the umbrella shrinks and relaxes constantly, and the water pressure in the umbrella cavity is pushed out of the body, allowing it to swim slowly in the opposite direction. Other marine creatures such as octopuses and squid also swim by spraying seawater, which is similar to the microrobot driven by the jet flow generated via acoustically actuated bubbles. Hence, several bubble-driven bionic microrobots and microswimmers based on acoustic and magnetic fields have been manufactured, and their movement and operation abilities have been demonstrated in fluids [168]. However, in addition to imitating the motion characteristics of organisms, bubble-based bionic microrobots can combine real-time monitoring and imaging, intelligent motion control, image processing, and other technologies, making them more intelligent and further expanding their practical applicability in microfluidics.

In vivo microrobots based on bubbles

The ultimate goal of microrobots is to enter the human body and improve human health. Countless researchers are tackling the key technical problems. We can put forward a bold vision for the future development of bubble-based microrobots: clustered bubble microrobots are injected into the human body [208]; under excitation of the acoustic field and control of the magnetic field [164], or using ultrasound for in vivo micromanipulation [209], these bubble microrobots can access specific parts of the body; under the action of different excitation frequencies or external fields, they can perform drug delivery or surgical tasks [179]; meanwhile, several bubble-based microrobots simultaneously complete real-time volumetric optoacoustic tomography imaging and monitoring [210]; finally, these bubble microrobots collapse or dissolve in the body. A variety of artificial intelligence (AI) strategies can be combined to make them more intelligent and autonomous [150].

Microrobotics is a highly interdisciplinary field involving physics, chemistry, biology, and medicine. From theoretical exploration to practical application, innovative ideas and a thorough exploration by numerous researchers from different backgrounds are also required. Micro/nanobubbles have played a variety of functions and roles and are bringing more surprises to researchers. We expect that micro/nanobubbles will realize more important and diverse applications in the future.

## Figures and Tables

**Figure 1 micromachines-13-01068-f001:**
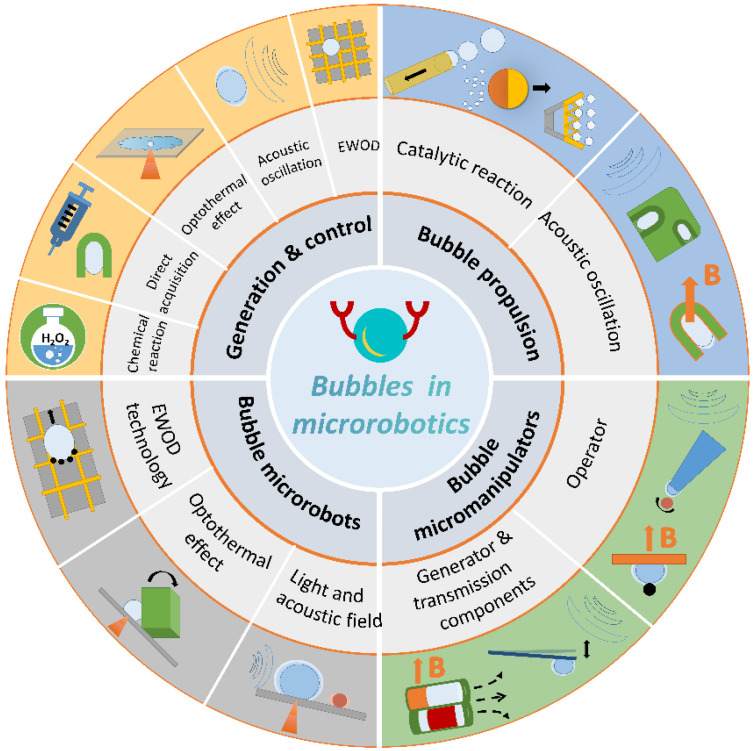
The generation and control methods of the bubbles and their roles in microrobotics.

**Figure 3 micromachines-13-01068-f003:**
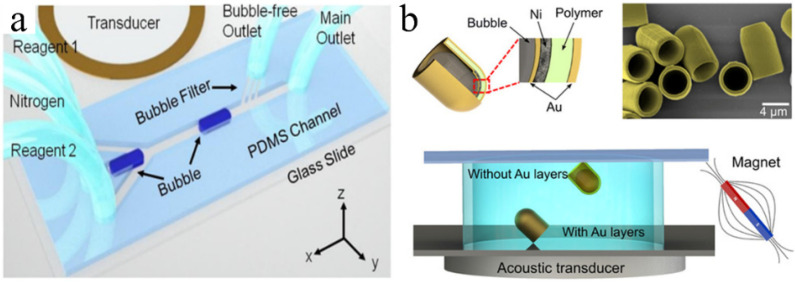
Bubbles generated via the direct acquisition method. (**a**) Bubbles generated by injecting nitrogen into the microfluidic device. Adapted from Orbay et al. [25] with permission from the Institute of Physics Publishing, Copyright 2017. (**b**) Air bubble trapped in a hydrophobic microcavity. Adapted from Ren et al. [33] with permission under the terms of the CC-BY-4.0 License, Copyright 2019.

**Figure 5 micromachines-13-01068-f005:**
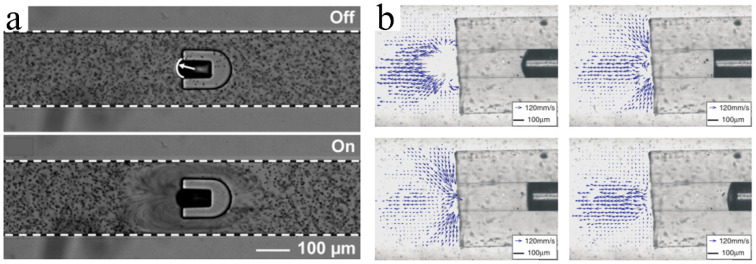
The flow field around oscillating bubbles excited by acoustic waves. (**a**) Acoustic streaming pattern around a bubble located in the horse-shoe structure of the microfluidic pipe. Adapted from Ahmed et al. [66] with permission from The Royal Society of Chemistry, Copyright 2009. (**b**) Acoustic streaming pattern around a bubble located in a tube closed at one end. Adapted from Dijkink et al. [70] with permission from the Institute of Physics Publishing, Copyright 2006.

**Figure 6 micromachines-13-01068-f006:**
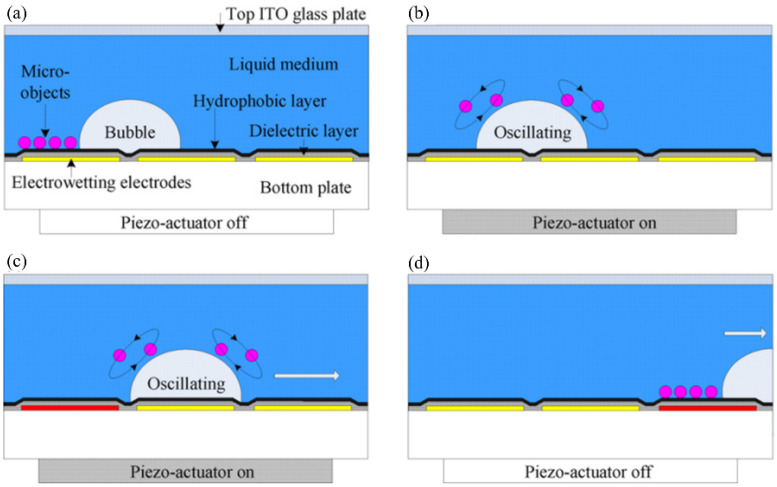
Oscillating bubble micromanipulation process driven by EWOD technology and acoustic waves. (**a**–**d**) Capturing, carrying, and releasing of objects via an oscillating mobile bubble. Adapted from Chung et al. [74] with permission from the Institute of Physics Publishing, Copyright 2012.

## Data Availability

Not applicable.

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
