# Peer review of "Review of Bubble Applications in Microrobotics: Propulsion, Manipulation, and Assembly"

_micromachines, 2022, doi:10.3390/mi13071068_

Round 1
Reviewer 2 Report
This review summarizes recent advances on bubble related microrobots. There were many review papers about the studies of bubble propulsion or bubble mediated manipulation previously. In this manuscript, the authors have managed to present those studies as a bigger group, which is helpful for researchers specially working on microbubbles. However, a bigger group brings more distraction and less interests. The manuscript could interest researchers working on various microrobots in a more efficient way after some essential improvements.
1. The purposes of using bubbles in microrobots are highly different. For example, the acoustic microrobots use bubbles as on-chip actuators while the chemical jets simply exhaust bubbles for achieving reaction force. Considering this huge difference, the authors should explain better the reason for putting them together. This reason should not be just the apparent fact that they all use bubbles. It should be why they use bubbles, i.e. the advantageous properties of bubbles.
2. The authors first introduces the generation of bubbles then their applications. I suggest to add one paragraph in each section to summarize the necessity and advantages of each technique or application.
3. The classification logic of the referred papers in some paragraphs should be improved. For example, Section 3.1 is classified according to the geometry of the microrobots (particle, tubular and micromachine). This leads to an embarrassing situation that any other shape than the particle and tube will be classified as a micromachine. I cannot tell the scientific reason for this classification. The classification of 3.1 needs to be re-organized unless the geometry does influence the function or physics of those bubble-propelled microrobots a lot. However, I have not seen the related text.
Some paragraphs, for example, from line 489 in page, are arranged to describe papers classified according to research groups. I understand that studies from one group are continuous and it is probably more effortless to write one paragraph for one group. However, the classification standard should be based on a common point on science. I notice that the authors classify the microrobots according to the geometry in 3.2. I think here is quite reasonable because the geometry does play an important role on the locomotion mode of acoustic microrobots, different from the chemical jets.
4. The authors should talk about more application limitations of each kind of microrobots. Some of them have been long-standing issues, for example, the absence and harm of H2O2 in the body.
5. I suggest the authors to add a concept figure as Figure 1 as an overall summary of the fabrications, applications, advantages and disadvantages of bubble-related microrobots.
